# Double Network Physical Crosslinked Hydrogel for Healing Skin Wounds: New Formulation Based on Polysaccharides and Zn^2+^

**DOI:** 10.3390/ijms241713042

**Published:** 2023-08-22

**Authors:** Shenghao Cui, Faming Yang, Dingyi Yu, Chao Shi, Di Zhao, Liqi Chen, Jingdi Chen

**Affiliations:** Marine College, Shandong University, Weihai 264209, China; 202000810226@mail.sdu.edu.cn (S.C.); 202121023@mail.sdu.edu.cn (F.Y.); 202000810156@mail.sdu.edu.cn (D.Y.); 201936673@mail.sdu.edu.cn (C.S.); 201900810068@mail.sdu.edu.cn (D.Z.); 201900810011@mail.sdu.edu.cn (L.C.)

**Keywords:** marine polysaccharide, semi-dissolved acidified & sol-gel conversion method, antibacterial properties, wound healing

## Abstract

Developing convenient, efficient, and natural wound dressings remain the foremost strategy for treating skin wounds. Thus, we innovatively combined the semi-dissolved acidified sol-gel conversion method with the internal gelation method to fabricate SA (sodium alginate)/CS (chitosan)/Zn^2+^ physically cross-linked double network hydrogel and named it SA/CS/Zn^2+^ PDH. The characterization results demonstrated that increased Zn^2+^ content led to hydrogels with improved physical and chemical properties, such as rheology, water retention, and swelling capacity. Moreover, the hydrogels exhibited favorable antibacterial properties and biocompatibility. Notably, the establishment of an in vitro pro-healing wound model further confirmed that the hydrogel had a superior ability to repair wounds and promote skin regeneration. In future, as a natural biomaterial with antimicrobial properties, it has the potential to promote wound healing.

## 1. Introduction

The regenerative capacity of skin depends on the speed and quality of wound healing. However, bacterial infections can disrupt this orderly process, forming chronic wounds and scars [1,2]. Many dressing therapies have been developed to promote wound healing while protecting the wound from bacterial infection [3,4,5,6]. Among these options, a double network hydrogel consisting of two cross-linked networks with distinct physical properties emerges as a highly promising approach for crafting wound dressings. Crosslinking tight, rigid networks provide the “rigidity” of hydrogels, and crosslinking loose, flexible networks maintain the “flexibility” of hydrogels, which can effectively harvest their advantages by adjusting the ratio of the two [7]. However, it still exhibits inadequate mechanical properties and low self-healing properties [8], while the use of chemical cross-linking agents poses a significant safety risk. Hence, it is particularly important to develop a physically cross-linked double network hydrogel that is safe and antibacterial and promotes healing, thus facilitating its application in tissue repair.

The advancement of physical cross-linking techniques (such as electrostatic interactions, hydrogen bonding, ionic cross-linking, etc.) [9] has endowed hydrogels with commendable mechanical and self-healing attributes owing to the existence of weakly reversible non-covalent bonds. For example, Rita et al. made a physically cross-linked double-net hydrogel with high strength and toughness by electrostatic bonding and ionic cross-linking [10]. Furthermore, addressing the issue of hydrogel aggregation or precipitation during preparation remains a prominent research focus. The semi-dissolved acidified sol-gel method used by Zhao et al. [11] and Kim et al. [12] is a good solution. The principle involves gradually dissolving alkaline metal compounds in acids to release metal ions progressively. These ions serve as cross-linking agents that react with other raw materials, culminating in the creation of hydrogel systems. Nonetheless, the frequently employed acid in this method is acetic acid, necessitating an extra washing step to eliminate residues. Hence, based on safety and economic considerations, exploring alternative strategies with low toxicity is necessary.

The successful preparation of physically cross-linked double-net hydrogels depends on the conformational compatibility of their constituents. Natural polysaccharides from marine sources have been a hot topic of research. For example, sodium alginate (SA) and chitosan (CS) [13,14] are widely used for the preparation of functional hydrogels due to their safety, favorable bioactivity, and degradability [15]. Meanwhile, the ionic cross-linking between the α-L-glutamine (G unit) structure of SA and divalent cations, resulting in ‘egg box’ model-structured hydrogels [16], does not fully satisfy practical mechanical requirements. However, CS, as a cationic polyelectrolyte, is not only widely used in tissue regeneration due to its good antibacterial and anti-inflammatory functional activity [17]. It can also physically be cross-linked with SA through electrostatic interaction to form hydrogels with better healing effects [18].

Most importantly, incorporating metal ions into hydrogel systems promotes the formation of hydrogel systems and provides more excellent biological properties. Zn is an indispensable trace element in the human body and a vital metal substance in modern industry. Zn has a wide range of applications in the field of materials, such as zinc-based nanocomposites [19], active electrodes [20], batteries [21], photocatalysts [22], polymer nanocarriers [23], wound dressings, etc. In wound dressings, Zn can promote wound healing by inducing skin regeneration and eradicating bacteria and has been widely used as a cross-linking agent and functional substance [24,25].

Therefore, we replaced the conventional acetic acid with gluconolactone (GDL) as a solubilizing agent for alkali metal compounds and prepared a marine polysaccharide-based physical double network crosslinked hydrogel (SA/CS/Zn^2+^ PDH) by semi-dissolution acidification sol-gel conversion method. The COO^−^ of SA and the -NH_3_^+^ of CS are combined by electrostatic interaction as the first network to maintain the hydrogel skeleton and enhance the elasticity of the hydrogel. Zn^2+^ is gradually dissociated by protonated basic zinc carbonate with gluconolactone (GDL), making it easier to crosslink with SA ions as a second network to dissipate energy and improve the mechanical properties of hydrogels. FTIR showed a significant redshift, indicating that two different physical cross-linked networks can be cross-linked by hydrogen bonds. Subsequently, we conducted a series of evaluations on its physical and chemical properties, mechanical properties, antibacterial ability, biocompatibility, and mouse cortical wound healing effect. The aim is to develop a new method for the preparation of environmentally friendly physical hydrogels with significant antibacterial properties based on natural polysaccharides and to apply them in wound repair.

## 2. Results and Discussion

### 2.1. Design Strategy for SA/CS/Zn^2+^ PDH

In this paper, we proposed a new preparation method (Figure 1A). Diverging from the conventional direct dissolution of basic zinc carbonate in acetic acid solution, our approach involves the uniform mixing of three components, namely basic zinc carbonate, SA, and CS, in a solution to create a slurry mixture. Subsequently, GDL is introduced into the system to enable continuous hydrolysis and proton release to dissolve basic zinc carbonate. During the ongoing stirring process, CS undergoes gradual dissolution. Basic zinc carbonate is gradually dissociated by protonated amino acids to release zinc ions. Within the SA/CS system, the first flexible network is formed by the electrostatic interaction between the cation-NH3 and the anion COO^−^ to maintain the skeleton structure of the hydrogel. Simultaneously, the SA chain was more stable and ionically crosslinked with the gradually released Zn^2+^ to form a SA/Zn^2+^ second rigid network, which was used as a sacrificial layer to provide the rigidity required by the hydrogel. This straightforward synthesis approach circumvented uneven precipitation and the overabundance of rigid network cross-linking. Such factors can otherwise result in heightened brittleness of the hydrogel. Furthermore, our method permits the creation of physically cross-linked double network hydrogels possessing controllable size and shape, achieved through electrostatic interactions and ion cross-linking.

The sol-gel transition was confirmed by the vital bottle tilting method (Figure 1B). The presence of Zn^2+^ greatly shortens the gel-forming time, and with the increase of Zn^2+^ concentration, the gel forming time is shorter. This is because the introduction of Zn^2+^ increases the SA/Zn^2+^ second hydrogel network, and this interpenetrating double network structure greatly shortens the hydrogel molding time. The gelation time of SA-CS-30 was 5 min, much lower than that of SA-CS-0 (122 min), which was expected to be used for the in-situ formation of hydrogels. This phenomenon occurs because Zn^2+^ was essentially hydrophilic, and its addition increased the concentration of other polymer solutions. This increased the chance of cross-linking CS with SA and shortened the gelation time.

### 2.2. Chemical Structure and Property Analysis

#### 2.2.1. Surface Appearance

The hydrogel fibers that reached the dissolution equilibrium were lyophilized, and the cross-section and surface morphology of the hydrogel fibers were observed by SEM (Figure 2). The hydrogels that reached solubilization equilibrium were lyophilized, and their surface morphology and microscopic pore structure were observed by SEM (Figure 2A). As shown in Figure 2A, the porous 3D meshwork of the SA/CS/Zn^2+^ PDH hydrogels would provide an ideal habitat for exudate uptake and cell growth. At the same time, due to the strong skeleton structure, it will only swell after absorbing enough water but will not break. Figure 2B showed that the introduction of Zn^2+^ significantly reduced the pore size of the hydrogel. This is because Zn^2 +^ and free SA formed a second network structure of the hydrogel through ion cross-linking, which increased the degree of cross-linking of the hydrogel, leading to a decrease in pore size and a denser skeleton.

#### 2.2.2. FTIR Analysis

The FTIR spectra of the SA, CS and SA/CS/Zn^2+^ PDH are shown in Figure 3A. They have similar infrared absorption. For example, the stretching vibration of C = O bond and the bending vibration of the N-H bond produce characteristic absorption peaks near 1500 cm^−1^; while the bending vibrations of O-H bonds form characteristic absorption peaks around 3300 cm^−1^. Notably, the C = O stretching vibration of SA and CS formed characteristic absorption peaks near 1593 cm^−1^ and 1612 cm^−1^, respectively, while an obvious red shift occurred in SA/CS/Zn^2+^ PDH, and the characteristic absorption peak appeared at 1564 cm^−1^. Likewise, the C-O stretching vibration of SA and CS formed characteristic absorption peaks near 1021 cm^−1^ and 1089 cm^−1^, respectively, while the characteristic absorption peak appeared at 1016 cm^−1^ in the SA/CS/Zn^2+^ PDH system. This phenomenon could be attributed to the interpenetrating cross-linking between the two double network structures through many hydrogen bonds [26]. Most importantly, a comparison of Figure 3A(i–iii) showed that no new characteristic peaks of SA/CS/Zn^2+^ PDH were found to appear due to the increase of Zn ion concentration. No bond-breaking recombination occurred, thus confirming that the double-net hydrogels obtained by the semi-soluble acidification and sol-gel transformation methods were achieved exclusively by physical cross-linking.

#### 2.2.3. Thermal Stability of Hydrogels

The hydrogel experiences two major weight loss processes during the rise from 40 °C to 800 °C (Figure 3B). First, starting from 40 °C, the water in the hydrogel undergoes evaporation, leading to an approximate 8% reduction in hydrogel weight by 100 °C. Subsequently, thermal decomposition occurs smaller molecules situated on the surface of the double network hydrogel undergo decomposition at lower temperatures, whereas larger molecules decompose at elevated temperatures [27]. As illustrated in the figure, the weight loss in this stage was larger, and the most weight loss was achieved with a final mass retention rate of about 25%. The average temperature at which water evaporation occurred in the Zn^2+^-containing groups exceeded that of the blank gel group. This is because the introduction of Zn^2+^ made the free SA in the system form an eggshell-like second network structure, which acted as a ‘sacrificial bond’ to make the hydrogel more thermally stable [28]. The SEM analysis (Figure 2) showed that the groups containing Zn^2+^ have a more compact three-dimensional network structure, which supported this view.

#### 2.2.4. Swelling Behavior of Hydrogels

To assess the swelling, hydrogels were incubated in 0.9% sodium chloride solution. Observation Figure 4A showed that all groups maintained a similar trend and regularly swelled to equilibrium. The average selling rate of the group without Zn^2+^ was significantly higher than that of the three groups with Zn^2+^. The swelling ratio of DN hydrogels decreased with increased Zn^2+^ content: After 10 h, SA-CS-10 reached equilibrium at about 1500%, while SA-CS-30 only reached equilibrium at 500%. Interestingly, the former showed a rapid water absorption rate (exceeding 500% within 1 h), while the latter rapidly reached a stable level (approximately 4 h). An increase in Zn^2+^ concentration leads to higher crosslinking density, reduced exposure of the internal polymer chain to water molecules, shorter time to reach equilibrium, and lower swelling degree of the hydrogel [29]. Another contributing factor is that Zn^2+^ is competitively chelated with CS, leading to the formation of a new network structure. Consequently, the free SA content increased and partially dissolved in water during the swelling process, which influenced the hydrogel’s composition and resulted in a decreased swelling degree.

#### 2.2.5. Water Retention Property

Comparing the water content under lyophilization and drying conditions and the water retention curves (Figure 4B–D). The water content of the hydrogels decreased with the increase of Zn^2+^ content. The increase in crosslinking density made the three-dimensional network skeleton of hydrogel more compact, leading to decreased water content. However, the water retention rate was positively correlated with Zn^2+^ content, which again proved the close relationship between Zn^2+^ content and crosslinking density. The tight network skeleton structure made the water not easy to lose. It improved the water retention performance of the hydrogel, which contributed to wound healing according to the wet healing theory [30].

### 2.3. Mechanical Strength and Rheology

During wound healing, hydrogel dressings must withstand the forces of surrounding skin and tissues and respond to sudden changes in the external environment [31]. Therefore, it is necessary to investigate the dynamic rheological properties of hydrogels. The G′ denotes the ability of the hydrogels of storing elastic deformation energy and represents the stiffness of hydrogels. It can be seen from Figure 5A that in the whole scanning range, the storage modulus (G’) was always greater than the loss modulus (G″), and the elastic behavior was dominant relative to the viscous behavior. This indicated that the change of frequency would not destroy the structure of the hydrogel, and the gel was stable and exhibited viscoelastic hydrogel. G′ is proportional to the content of Zn^2+.^ This indicates that introducing Zn^2+^ increased the crosslinking density and enhanced the deformation resistance. As shown in Figure 5B, the compressive modulus of the hydrogel increased, and the compressive strength increased with the increase of Zn^2+^ concentration, which is consistent with the rheological experimental results. Among them, the SA-CS-30 group reached 48 KPa, 26 times the compressive strength (1.8 KPa) of the SA/CS single network hydrogel (SA-CS-0 group). This indicates that the addition of Zn^2+^ improved the elasticity of the hydrogel and can better cope with the complex external environment. In addition, the polysaccharide-based physical double-network crosslinked hydrogels prepared by the semi-dissolution acidification sol-gel conversion method in this work have higher G’ than the same SA/CS-based hydrogels. The G’ of SA/CS/ZnO hydrogel prepared by Zhang et al.is about 10^3^ Pa [32], and the G’ of SA/A-HA (aldehyde hyaluronic acid)/Zn^2+^/CMCS (carboxymethyl chitosan) hydrogel prepared by yan et al.was below 10^3^ Pa [33], while the G’ of the hydrogel in this work was between 10^4^ and 10^5^ Pa.

SA gel lacks strong enough mechanical properties, significantly limiting its practical application. If the fibers are weak, they can be easily damaged during application. It can be seen from the texture profile analysis (TPA) of hydrogels (Figure 5C–F) the hardness and cohesion of the hydrogel increased with the increase of Zn^2+^ content. The results of TPA further proved that with the increase of Zn^2+^ content, the degree of crosslinking of hydrogels also increased, which enhanced the mechanical strength.

Figure 5C–F shows that the dominant behavior was dominant for all groups relative to the viscous behavior, showing gel-like behavior. With the increase of Zn^2+^ concentration, the crosslinking degree and stiffness of the hydrogel increased, and the mechanical properties increased. This is due to the increased Zn^2+^ content, which improved the integrity of the second network and enhanced the overall crosslinking of the hydrogel. The rheological behavior of SA/CS/Zn^2+^ PDH suggested a suitable application in wound healing.

### 2.4. In Vitro Activity of Hydrogels

#### 2.4.1. In Vitro Release of Zn^2+^

Zinc plays an important role in wound healing, and the content of Zn^2+^ significantly affects the biological activity of hydrogels [34]. We infiltrated SA-CS-10, SA-CS-20, and SA-CS-30 in a PBS buffer to observe the release of Zn^2+^ within one day. It can be seen from Appendix A that the Zn^2+^ release rate of each component reached more than 70% in the first 6 h, and the Zn^2+^ release rate tended to be balanced after 6 h. This may be because the hydrogel reached swelling equilibrium at about 10 h, and the release rate of Zn^2+^ slowed down. In addition, at the same time, the group with high zinc ion content showed a lower release rate, which was closely related to the degree of cross-linking. The higher the degree of crosslinking, the more difficult it is for Zn^2+^ to break away from the original hydrogel network system.

#### 2.4.2. Antioxidant Activity of Hydrogels In Vitro

Elevated levels of reactive oxygen species (ROS) can lead to oxidative damage in cells, which is detrimental to skin wound healing [35]. Zinc is one of the important components of the human body ‘s antioxidant system. Introducing an appropriate amount of Zn^2+^ can effectively improve the antioxidant activity of the material [36]. The antioxidant activity of CS/CS/Zn^2+^ PDH was evaluated by DPPH free radical scavenging method. As shown in Appendix A, each hydrogel component exhibited a certain antioxidant effect. The introduction of Zn^2+^ into the system significantly improved the antioxidant effect (up to 10%). Although the release rate of the SA-CS-30 group was low, due to the large concentration of Zn^2+^, more Zn^2+^ could be released at the same time, which was useful for higher antioxidant capacity. Compared with the antioxidant capacity of 5% VC, the hydrogels of each component were slightly insufficient. The reason is that the content of Zn^2+^ in the prescription was relatively low, and it was difficult for Zn^2+^ to break away from the original hydrogel system in a short time. Hence, the concentration of Zn^2+^ in the PBS buffer was low. Nonetheless, CS/CS/Zn^2+^ PDH can still accelerate the wound-healing process to a certain extent.

#### 2.4.3. Antimicrobial Property of Hydrogels

The antibacterial properties of the hydrogels were evaluated using the disc diffusion method. (Figure 6A). SA/CS/Zn^2+^PDH showed strong inhibition against *E. coli* and *S. aureus.* The hydrogel without Zn^2+^ had little inhibitory effect on *E. coli*, and when Zn^2+^ was added, its inhibitory effect was significantly increased, and the content of Zn^2+^ was proportional to the antibacterial effect (Figure 6B), indicating that the hydrogel achieved antibacterial effect without using antibiotics, and has excellent potential as a skin wound dressing. The electrically charged amino group of chitosan can disrupt the bacterial cell wall, resulting in the release of intracellular fluid [37]. However, the SA-CS-0 group showed weak antibacterial properties, possibly due to a higher cross-linking between CS and SA through electrostatic interaction. This reduced availability of free positively charged chitosan. Moreover, Zn^2+^ can adsorb negatively charged bacteria and react with them to destroy their structure, thus playing a sterilizing role. After the bacteria are killed, Zn^2+^ will fall off and adsorb the remaining bacteria again, thus having a repeated sterilization effect [38]. After the introduction of Zn^2+^, the Zn^2 +^ competitively combined with SA to expose freer CS, and the excess Zn^2+^ cooperated with the free CS to exert a stronger antibacterial effect. As seen in Figure 6B, SA-CS-20 showed stronger antibacterial activity at the same Zn^2+^ concentration. Therefore, the presence of Zn^2+^ can form a protective film on the material’s surface to affect the proliferation of *E. coli* and *S. aureus* while controlling their degradation rate in the desired range [39].

### 2.5. In Vitro Evaluation of Cytocompatibility

Biocompatibility is a crucial characteristic of hydrogel dressings worth exploring [40]. CCK-8 activity assay and AO & EB staining were used to detect the toxicity of the hydrogels. It can be seen from Figure 7A that each constituent hydrogel was non-toxic to HaCaT cells within the experimental concentration range, and the introduction of Zn^2+^ could significantly increase cell viability, which agreed with the results reported in previous articles [41]. The SA-CS-20 component showed the highest cell viability, possibly due to the high concentration of Zn^2+^ which played a certain inhibitory effect.

After incubating the cells in the hydrogels, they were stained with AO and EB for live and dead cell detection (green fluorescence for live cells; red fluorescence for dead cells), followed by observation using confocal microscopy. AO/EB staining results once again demonstrated the non-toxicity of each component hydrogel. Figure 7B shows the cells survived well in all hydrogels, and no red dead cells were observed, which verified the good.

### 2.6. In Vivo Activity Evaluation

#### 2.6.1. In Vivo Wound Healing in a Total Skin Defect Model

Wound recovery on days three, five, seven, and nine was observed, and the analysis of Figure 8 indicated that the Zn^2+^-containing groups exhibited superior back wound healing compared to the blank group overall. The quantitative statistics of wound size recovery showed that the grafted wound areas were all reduced, and the rate of wound skin contraction was accelerated with the increase of Zn^2+^ concentration. The groups with Zn^2+^ levels higher than 5 mM on day three were statistically different from the blank group (*p* < 0.05). Compared with the control group, on day five, wound recovery was significantly improved in the Zn^2+^-containing groups, and there was still a statistical difference between the SA-CS-20 and SA-CS-30 groups and the blank group (*p* < 0.05). On day seven, the rate of healing in the Zn^2+^-containing groups significantly increased compared to the positive control group, and there was a significant difference between the SA-CS-20 and SA-CS-30 groups and the blank group (*p* < 0.05). On day nine, the wound surface of the Zn^2+^-containing groups had healed substantially without significant depressions, and a thicker epidermis covered the wound surface. However, the group without Zn^2+^ displayed more depression than the blank group, and the central part of the wound appeared sunken. A dry and thin crust was observed on the wound surface, and a small amount of exudation was still visible after removing the crust, with the epidermis still partially failing to cover the wound.

Preliminarily, it has been demonstrated that the hydrogel with the addition of Zn^2+^ had a strong healing-promoting ability [42], and the healing-promoting ability increased gradually with the increase of Zn^2+^ content.

#### 2.6.2. Histological Evaluation of Regenerated Tissue

After conducting the above experiments, the hydrogel’s effective wound healing-promoting ability was preliminarily established, and further experiments were conducted for verification.

The results of H&E staining are shown in Figure 9. The Zn^2+^-containing groups contained fewer inflammatory factors on day three compared to the negative control group and SA-CS-0 group. On day seven, the negative control group exhibited fewer inflammatory cells, the epidermis was incompletely healed, and the dermis contained fewer collagen fibers. In contrast, the Zn^2+^-containing groups showed no inflammatory reaction, and the hydrogel formed a tight and seamless interface with the peri-wound tissue. Moreover, many granulation tissues appeared, facilitating faster migration of cells from the peri-wound tissue [43]. By day 14, the cells were completely infiltrated in the hydrogel, and a few hair follicles appeared in the Zn^2+^-containing groups. The repair of the epidermis and dermis was similar to that of normal skin [44]. Overall, adding Zn^2+^ did inhibit the inflammatory response and further confirmed its good wound healing-promoting effect. And the inhibitory effect gradually increased with the increase of Zn^2+^ content, consistent with the wound healing experiment, and the ability to promote epithelial regeneration remained positively correlated with the Zn^2+^ content.

## 3. Materials and Methods

### 3.1. Experimental Materials

Chitosan (degree of deacetylation ≥ 95%, AR) and D- (+)-glucuronide δ-lactone (AR) were purchased from Macklin (Shanghai, China). Sodium Alginate (AR) and [ZnCO_3_]_2_- [Zn (OH)_2_]_3_) (AR) were supported by China National Pharmaceutical Group Corporation (Beijing, China). The strains of *Escherichia coli* (*E. coli*) and *Staphylococcus aureus* (*S. aureus*) were obtained from China Type Culture Collection Center. SPF weight 20 ± 2 g male mice were purchased from Hebei Fu (Beijing) Biotechnology Co., Ltd. (Beijing, China), production license number SCXK (Jing) 20190010. HaCaT cells were purchased from Suzhou Benacron Biotechnology Co. Acridine orange (AO)/ Ethidium bromide (EB) staining kit was purchased from Shanghai Biotech Biotechnology Co. (Shanghai, China). All chemicals were analytical degree reagents and used as received.

### 3.2. Preparation Method of SA/CS/Zn^2+^ PDH

SA/CS/Zn^2+^ PDH was prepared with reference to the modified Zhang et al. method [45]. The first was to make SA/CS/basic zinc carbonate form a uniform suspension system. Then, by adding a certain amount of GDL, the CS and basic zinc carbonate in the system were continuously dissolved, and finally the hydrogel system was formed. The detailed steps of which can be found in the Appendix A. In addition to the instructions, the tested hydrogel is 15 mm in diameter and 6 mm in thickness.

### 3.3. Gelation Time

The vial tilt method was used to estimate the time of gel formation [46]. A certain amount of GDL was added to the samples with different Zn^2+^ contents and stirred until the liquid flowed continuously, and the time was recorded. The detailed steps of which can be found in the Appendix A.

### 3.4. Physio-Chemical Properties of SA/CS/Zn^2+^ PDH

The surface morphology, chemical structure, and thermal stability of the SA/CS/Zn^2+^ PDH were characterized using scanning electron microscopy (SEM, JSM 6390, JEOL, Tokyo, Japan), Fourier infrared spectroscopy (FTIR, Shimadzu, Kyoto, Japan), and thermal analysis (TGA 449C, TA Instruments-Waters LLC, New Castle, DE, USA). The detailed procedures can be found in the Appendix A.

### 3.5. Swelling Property

The swelling property of SA/CS/Zn^2+^ PDH was determined according to previous reports [47]. Different samples were soaked in PBS buffer, and the swelling rate was calculated every 2 h. The detailed steps of which can be found in the Appendix A.

### 3.6. Water Retention Capacity

The water retention capacity of SA/CS/Zn^2+^ PDH was determined according to previous reports [48]. A certain amount of freeze-dried hydrogel was swollen in water and then dried in an 80 °C vacuum oven. The water retention rate was calculated by weighing at intervals. The detailed steps of which can be found in the Appendix A.

### 3.7. Mechanical Performance Testing

The texture profile analysis (TPA) of the hydrogel was performed using the XT Plus Texture Analyzer (Stable Micro Systems, Godalming, UK) [49]. According to the standard GB/T 16491-2008, the mechanical properties of the hydrogels were evaluated by uniaxial compression experiments using a universal test machine (HounsefieldH25K, UK) [50]. Frequency scan experiments were conducted using an Anton Pa1-100 rad with a strain of 1%, utilizing an MCR101 rheometer (Anton Paar, North Ryde, Austria) with a parallel plate (Ø50 mm) [51]. The detailed procedures can be found in the Appendix A.

### 3.8. In Vitro Release of Zn^2+^

The in vitro release of Zn^2+^ of SA/CS/Zn^2+^ PDH was determined according to previous reports [52]. The content of Zn^2+^ in the hydrogel was determined by inductively coupled plasma mass spectrometry (ICP-MS) (Thermo Scientific, XSERIES 2, Waltham, MA, USA). The detailed steps of which can be found in the Appendix A.

### 3.9. Antioxidant Activity of SA/CS/Zn^2+^ PDH In Vitro

The DPPH radical scavenging activity of SA/CS/Zn^2+^ PDH was measured using the technique reported before [53]. The detailed steps of which can be found in the Appendix A.

### 3.10. In Vitro Evaluation of Antibacterial Properties

The inhibition activity of the hydrogels against *E. coli* and *S. aureus* was evaluated using the disc diffusion method [54]. The detailed steps of which can be found in the Appendix A.

### 3.11. Biocompatibility Testing

Biocompatibility of SA/CS/Zn^2+^ PDH was assessed by HaCaT cells. Cytotoxicity was determined by AO/EB staining [55] and CCK-8 assay [56]. The detailed steps of which can be found in the Appendix A.

### 3.12. In Vivo Wound Healing Test

This study was carried out in compliance with the approved protocol by the ethics committee of Shandong University and its management regulations. The wound healing-promoting capability of SA/CS/Zn^2+^ PDH was assessed using a mouse model of full-thickness cortical injury. Wound healing rates were quantified, and wound remnants were visualized using Image J software (version 2.3.0). H&E and Masson’s staining were conducted on the regenerated skin at the wound site on days 3, 7, and 14 to evaluate the healing outcome.

### 3.13. Statistical Analysis

Experiment data were reported as the mean ± standard deviation (mean ± S.D.). Analysis was carried out using SPSS software (version 25). The LSD procedure was used, the significance level was set at *p* < 0.05, and multiple comparisons were made between the groups. The significance between groups is marked by letters: the same letters represent no significant difference between groups, and the different letters represent significant differences between groups. Capital letters indicate *p* < 0.01, lowercase letters indicate *p* < 0.05. “*” represents *p* < 0.05.

## 4. Conclusions

This paper describes the successful preparation of a safe and eco-friendly physical double network hydrogel dressing through a semi-dissolution acidification sol-gel method, employing SA, CS, and basic zinc carbonate as raw materials for therapeutic wound repair. The incorporation of a novel non-toxic acidifier, GDL, reduced the need for additional washing steps and prevented uneven precipitation during the hydrogel formation. The addition of Zn^2+^ significantly enhanced the mechanical properties of hydrogels (with a storage modulus ranging from 10^4^ to 10^5^ Pa) and imparted excellent antibacterial capabilities. Animal experiments demonstrated the hydrogel’s capability to effectively inhibit the inflammatory response, stimulate the formation of blood vessels and epithelial cells, and promote hair follicle regeneration, thereby showcasing its promising potential as a wound dressing. This straightforward preparation method for SA/CS/Zn^2+^ PDH hydrogel offers a novel strategy for designing and fabricating effective hydrogel dressings for skin wound treatment.

## Figures and Tables

**Figure 1 ijms-24-13042-f001:**
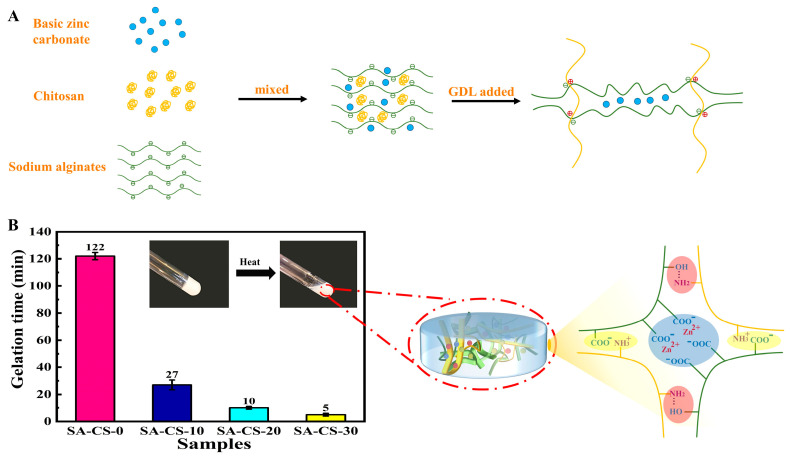
Forming mechanism and preparation of hydrogel. (**A**) Hydrogel preparation process. (**B**) Variation of gel time of hydrogels with different Zn^2+^ contents and crosslinking mechanism.

**Figure 2 ijms-24-13042-f002:**
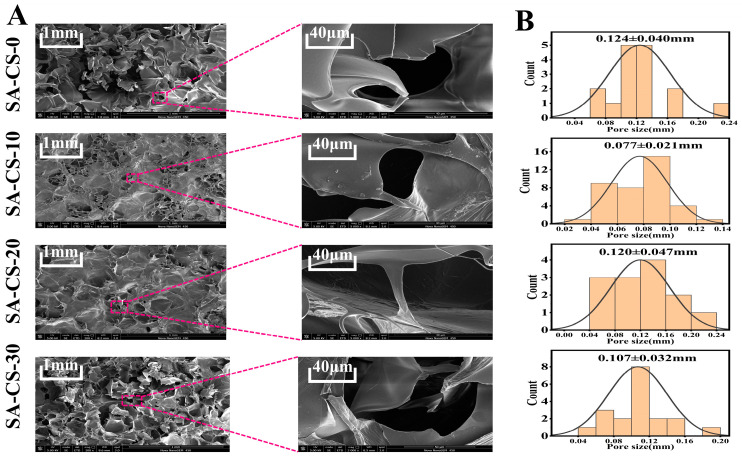
The surface appearance of DN gels with different Zn^2+^ contents. (**A**) SEM morphology of hydrogels with different Zn^2+^ contents. (**B**) The pore size of hydrogels with different Zn^2+^ contents.

**Figure 3 ijms-24-13042-f003:**
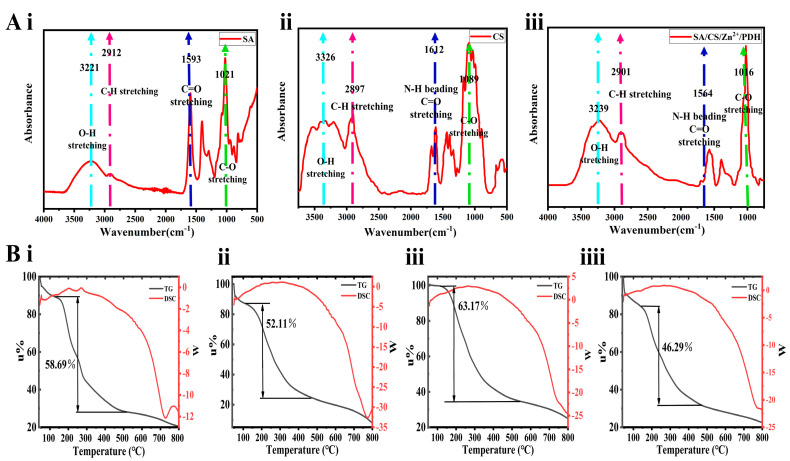
Characterization analysis of DN gels with different Zn^2+^ contents. (**A**) FTIR spectra of different components in this double network hydrogel. (**B**) TG-DSC curves of hydrogels with different Zn^2+^ contents. From left to right are the SA-CS-0 group, SA-CS-10 group, SA-CS-20 group, and SA-CS-30 group.

**Figure 4 ijms-24-13042-f004:**
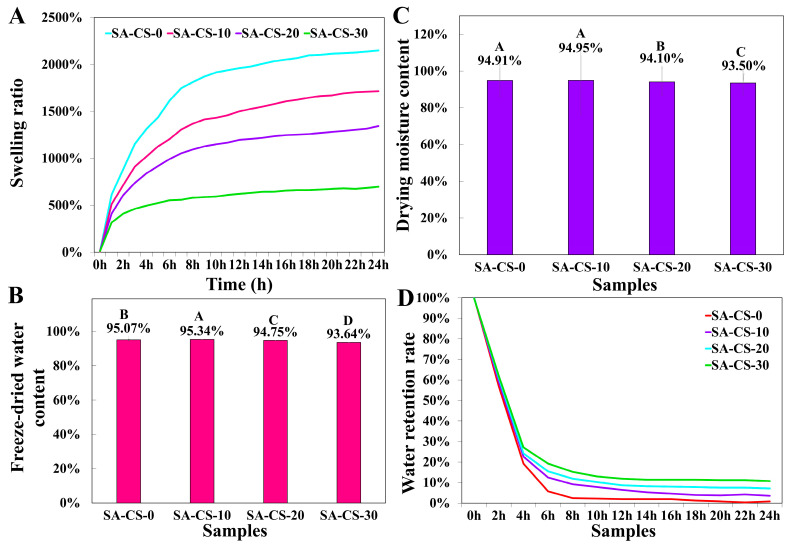
Evaluation of the swelling potential of this hydrogel. (**A**) The variation curves of the swelling ratio of hydrogels with different Zn^2+^ contents (n = 3). (**B**) The freeze-drying water content of hydrogels with different Zn^2+^ contents (n = 7). (**C**) The curves of drying water content of hydrogels with different Zn^2+^ contents (n = 7). (**D**) The variation of water retention ratio of hydrogels with different Zn^2+^ contents (n = 3).

**Figure 5 ijms-24-13042-f005:**
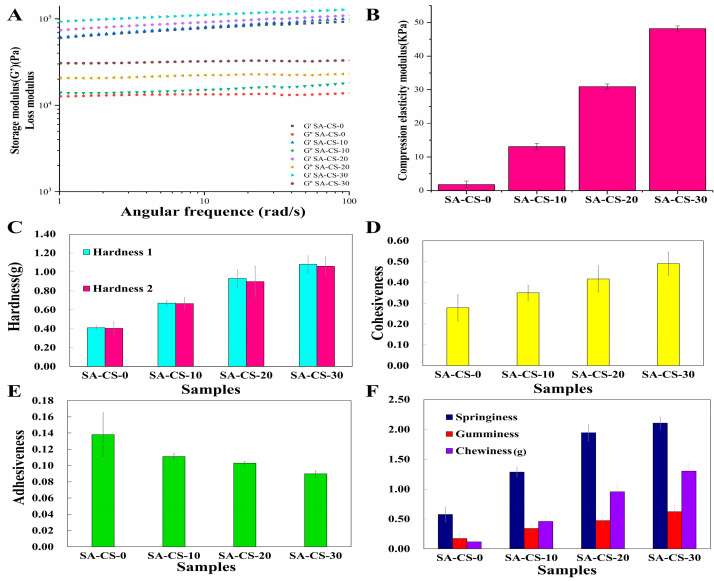
Evaluation of the mechanical and rheological properties of this hydrogel. (**A**) Frequency scanning rheological curve of hydrogel (n = 3). (**B**) Compression modulus of hydrogel (n = 3). (**C**) The hardness of the hydrogels with different Zn^2+^ content (n = 3). (**D**) The viscosity of the hydrogels with different Zn^2+^ content (n = 3). (**E**) The polymerization force of hydrogels with different Zn^2+^ content (n = 3). (**F**) Hydrogels’ elasticity, adhesive and chewing properties with different Zn^2+^ content (n = 3).

**Figure 6 ijms-24-13042-f006:**
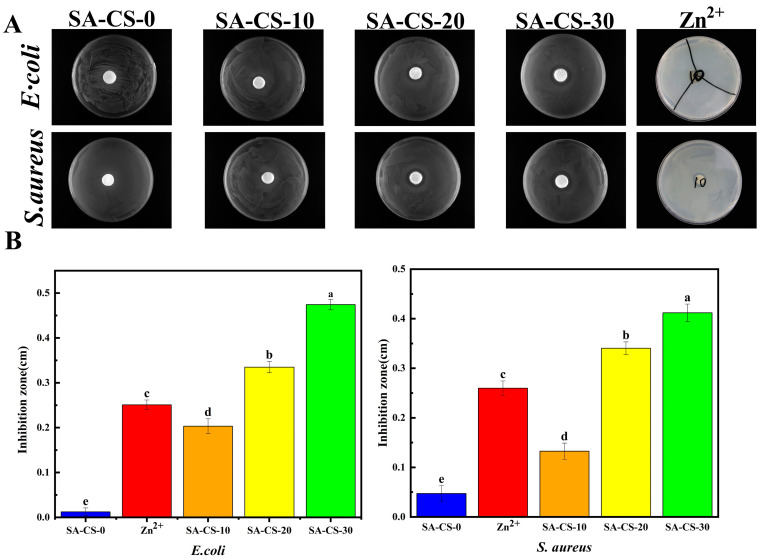
Evaluation of the bacterial inhibition effect of the dual network hydrogels with *E. coli* and *S. aureus* as examples. (**A**) Inhibition circles formed by the action of hydrogels with different Zn^2+^ contents. (**B**) Histogram of the radius of inhibition zone of hydrogels with different Zn^2+^ contents and 0.02 mol/L basic zinc carbonate (n = 3).

**Figure 7 ijms-24-13042-f007:**
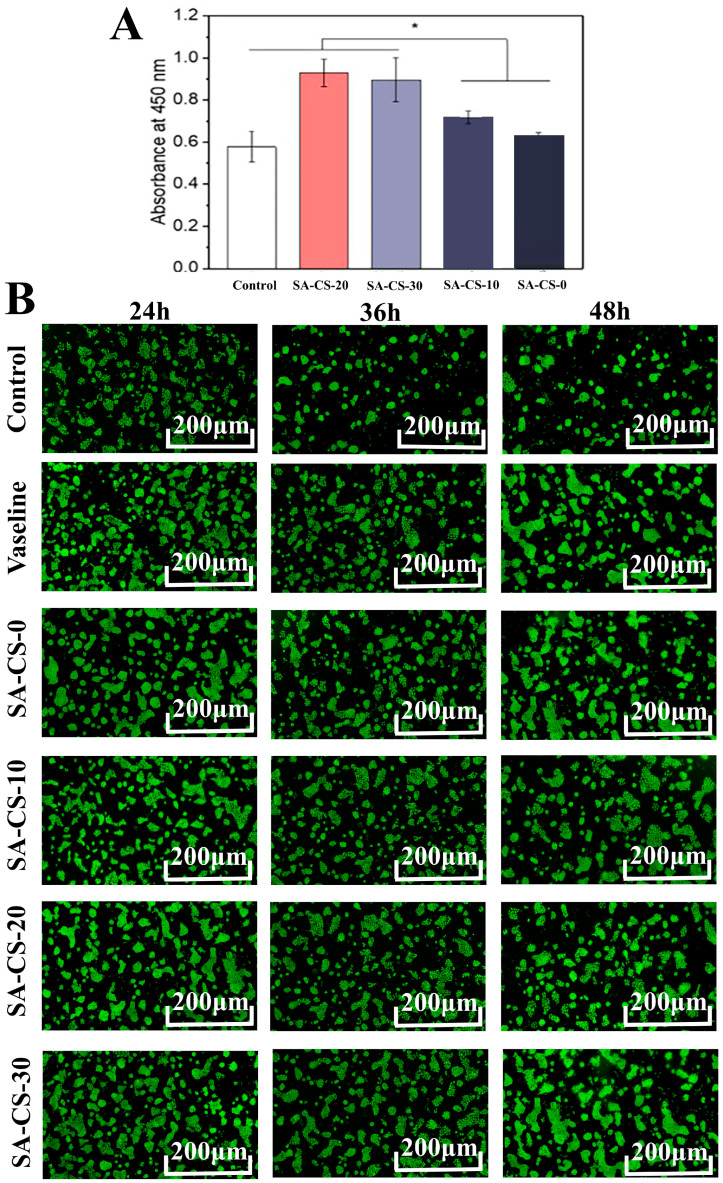
In vitro cytotoxicity test of this double network hydrogel. (**A**) Results of CCK-8 experiments with different components of hydrogel (n = 5). (**B**) AO/EB fluorescence staining for 24 h, 36 h and 48 h (magnification: ×200). AO fluoresces green after crossing the intact cell membrane and embedding in nuclear DNA. EB penetrates damaged membranes, interacts with DNA, and emits orange-red fluorescence.

**Figure 8 ijms-24-13042-f008:**
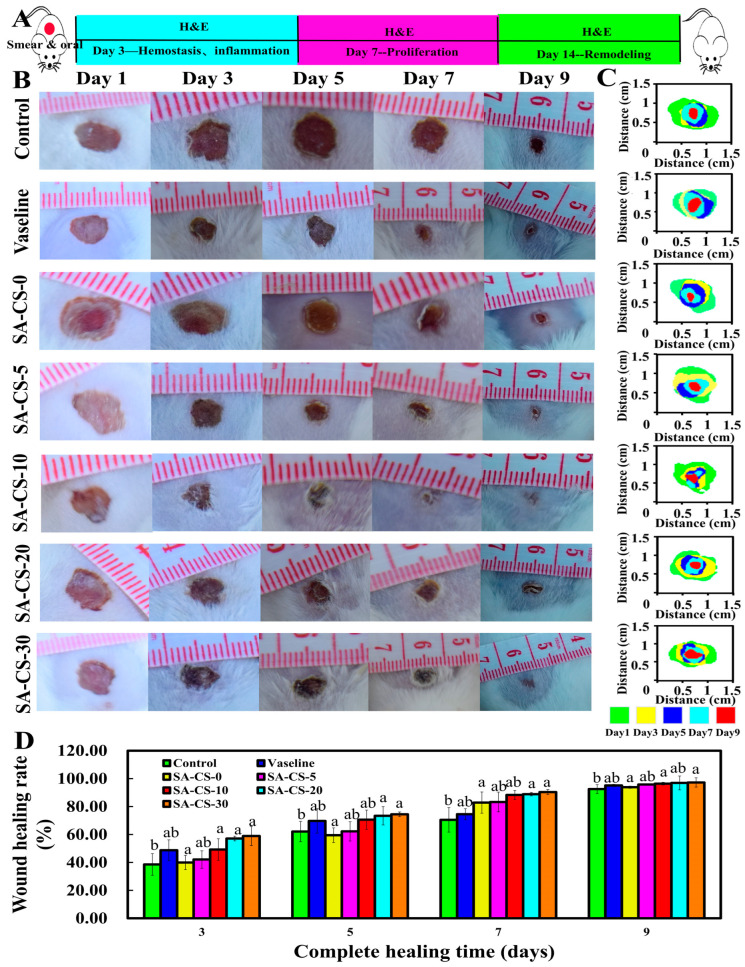
Evaluation of the in vitro pro-healing ability of this dual network hydrogel. (**A**) A simple flow chart of this in vitro experiment. (**B**) The physical effect of wound healing with different formulations of hydrogels. (**C**) Thermograms of wounds over time with different formulations of hydrogels. (**D**) Wound healing rates over time with different formulations of hydrogels.

**Figure 9 ijms-24-13042-f009:**
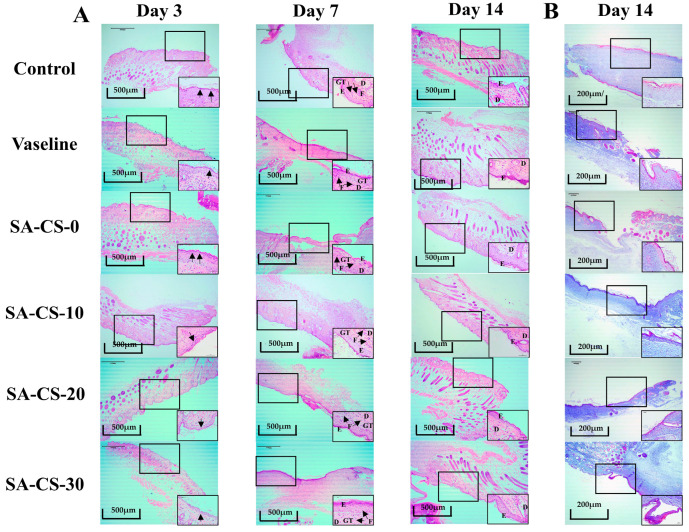
Representative images of H&E and Masson staining (4× and 40×, respectively). (**A**) H&E staining for each group on days three, seven, and 14. Black arrows on Day three indicate inflammatory cell infiltration, and on Day seven indicate fibroblast. Letters D, E, F and GT represent the dermis, epidermis, fibroblasts, and granulation tissue. (**B**) Masson staining for each group on day 14.

## Data Availability

Not applicable.

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
