# Peer review of "Double Network Physical Crosslinked Hydrogel for Healing Skin Wounds: New Formulation Based on Polysaccharides and Zn2+"

_ijms, 2023, doi:10.3390/ijms241713042_

Round 1

Reviewer 1 Report

I accurately reviewed the paper

Manuscript Number: ijms2535932

Title: Polysaccharide-based physical double-network crosslinked hydrogel prepared by semi-dissolved acidification sol-gel method for skin wound healing

submitted to International Journal of Molecular Sciences.

In this manuscript authors combined the semi-dissolved acidified sol-gel conversion method with the internal gelation method to prepare SA (sodium alginate)/CS (chitosan)/Zn2+  physically cross-linked double network (DN) hydrogels. The characterization results showed that with the increase of Zn2+ content, the hydrogels exhibited good physical and chemical properties, including rheology, water retention, and swelling properties. In addition, the hydrogels had good antibacterial properties and biocompatibility. Notably, the establishment of an in vitro pro-healing wound model further demonstrated that the hydrogel had a better ability to repair wounds and promote skin regeneration.

The topic is interesting, but the authors must solve some issues.

The title seems too long to me and I suggest to the authors:

Double network physical crosslinked hydrogel for healing skin wounds: new formulation based on polysaccharides and Zn2+

Introduction

The authors should expand the introductory part in order to allow readers to place this work within the current research landscape allowing to grasp the traits of advancement and originality. I suggest to better motivate the use of hybrids with Zn, giving an overview of its chemical and physical characteristics and its uses, also in composites materials. In my opinion, some important works in these fields should be cited and used for the above discussions.

a. Polymers 2023, 15(9), 2167; https://doi.org/10.3390/polym15092167 - 02 May 2023

b. Materials Research Express 1 (2014) 015040 DOI: 10.1088/2053-1591/1/1/015040

c. ACS Appl. Mater. Interfaces 2022, 14, 36, 40951–40958 https://doi.org/10.1021/acsami.2c10517

d. Polymers 2022, 14(21), 4484; https://doi.org/10.3390/polym14214484 - 23 Oct 2022

e. Journal of King Saud University Science 31 (2019) 398–411 https://doi.org/10.1016/j.jksus.2017.10.004

Results and Discussion

Did the authors verify in any way the actual amount or concentration of Zn2+ in the different preparations? An XPS study would be interesting.

The authors must explain the acronyms and insert a reference for the various tests performed (both of cytotoxicity tests and regarding all the assays).

Materials and methods

More information (purity and brand) must be provided about the reagents used, including solvents, and the equipment used (and only mentioned in the support).

The authors report the experimental part in the supporting section, also referring to previous works. However, in my opinion, a brief description must also be included in the main text to grasp the meaning of the work and make it easier to read.

Conclusion

The conclusions are very hasty. The authors must improve by highlighting the strengths and novelties of their work and leave an idea of the future perspective.

Figures

Figure 1A the authors have to correct "Galation time" on the y axis with Gelation time

Figure 2 B axis x must be corrected "aperture" with "pore size"

Figure 3 A must be corrected "Intensity (a.u.)" with Transmittance (%)

Figure 6 B the caption lacks self-consistency: what do the letters on the histograms refer to?

English requires corrections: there are typos and many sentences are too long.

In conclusion, the article is suitable for publication, but only after major revisions.

best regards

English requires corrections: there are typos and many sentences are too long.

Reviewer 2 Report

Cui and collaborators present a work on the preparation and characterization of a polysaccharide-based hydrogel having a double-network crosslinked structure promoted, internally, by ionic interactions between SA and CS and externally by the presence of zinc cations that induce the ionic gelation of SA. Then, different specimens prepared without Zn and with increasing amounts of metals were tested in skin wound healing applications, accompanied by cell viability and toxicity assays. Without a doubt, the authors present an enormous amount of information, but regularly, expressed in a repetitive manner and also confusing from time to time. Throughout the whole manuscript, there are some scientific arguments that need to be re-explained or supported by literature. Also, some experimental details must be incorporated to understand in a better way the results obtained from different characterization techniques. Regarding the format, the quality of all the figures must be increased, since in its current form it prevents the understanding of the explanations given by the authors. Also, several typos errors can be found along the whole manuscript. This work should pass through major revisions before being considered for publication.

Herein, I point out some comments that would help the authors to increase the quality of the manuscript.

1) line 28-29: "Among them, a double network hydrogel (DE) consisting of two 28 cross-linked networks with diverse physical properties is not a very promising strategy for the preparation of wound dressings" Is this what the authors wanted to say? I ask because, after that sentence, they even remark on other flaws of these materials so, as the first paragraph of the introduction, it is not helping at all to catch the attention of the readers.

2) line 35: it is missing the ")", probably after "etc."

3) line 67, the authors never define PDH

4) line 73: This is a good example of why authors need to improve the organization of the information. The paragraph begins and gives to the reader the idea of the importance of the electrostatic interactions within the system (CS-SA and SA-Zn), and suddenly, between they express "The two networks are cross-linked by hydrogen bonds". So, it would be very helpful to try to explain and create a more readable information, byrelating the different crosslinking mechanisms that can take place within the specimens

line 88: NH3 should be NH3

line 96 - 105: How can the authors ensure that during the gel preparation, CS is fully dissolved? The inset in Figure 1B reveal a whitish gel, typical for dispersed systems.

- In my opinion, SEM figures are very similar to each other and no clear trend can be extracted from them, how the authors calculated pore size distribution?

- FTIR analysis must be accompanied with a better Figure 3. None of the information written in the paragraph can be contrasted with the information given in Figure 3A. Also, the numbers in Figure 3B are very difficult to see.

- Both sections 2.2.3 and 2.2.4 are referred as "Thermal stability of Hydrogels"

- Regarding the thermal stability of hydrogels (section 2.2.3): "The average evaporation temperature of water in the group containing Zn2+ was higher than that in the blank group of gels, indicating that the joining of hydrophobic groups to each other after the addition of Zn2+ led to an increase in the ordered arrangement of water in the gels and higher thermal stability." I find it hard to agree with this argument considering the lack of experimental evidence. How can the authors argue about the order of the inner water? Also, i do not see how SEM analysis supports this statement.

- TGA measurements were performed on wet or dry samples?

- Figure 4 caption should be corrected by using Zn+2 notation

- I believe that in the swelling and water retention section, the authors also should discuss their results in terms of crosslinking degree between each sample (as they comment in the rheology results). The crosslinking density should also affect the swelling capacity of specimens

- line 209: Honestly, i consider that SEM results are not enough to state "consistent with the SEM results"

- Similar to other manuscript's sections, in the 2.4 section the author should try to unify the discussion and not just argue individually each observation. For example, you can expect a synergy in terms of the antimicrobial properties of these specimens, considering that both, the presence of Zn and the charged nature of the chitosan amino group should contribute to this property. Otherwise, it is not clear which one is the predominant. Also, can SA contribute to this property?

Regarding the last three sections, the information provided is enough and supports the possibility of using these materials for skin wound application. 

The manuscript turned out to be hard to read, with very repetitive sentences, and redaction and punctuation must be improved. In addition, there are a lot of typos and format mistakes 

Round 2

Reviewer 1 Report

The authors have responded adequately to the suggestions and the work has improved and is ready for publication.

best regards

Fair English

Author Response

Dear Professor,

Thank you very much!    Yours sincerely,

Faming Yang

Reviewer 3 Report

The authors responded to my suggestions. However, before publishing the article, I still recommend showing the FTIR spectra on the absorption scale instead of the transmittance (%), which is a calculated value.

Round 3

Reviewer 3 Report

The authors responded to my comments. The work can be published.